

# Preferences for achromatic horizontal, vertical, and square patterns in zebrafish (*Danio rerio*)

Lisa A. Rimstad[1,2], Adam Holcombe[1], Alicia Pope[1], Trevor J. Hamilton[1,2] and Melike P. Schalomon[1]

[1] Department of Psychology, MacEwan University, Edmonton, Alberta, Canada
[2] Neuroscience and Mental Health Institute, University of Alberta, Edmonton, Alberta, Canada

## ABSTRACT

The zebrafish (*Danio rerio*) is gaining popularity as a laboratory organism and is used to model many human diseases. Many behavioural measures of locomotion and cognition have been developed that involve the processing of visual stimuli. However, the innate preference for vertical and horizontal stripes in zebrafish is unknown. We tested the preference of adult zebrafish for three achromatic patterns (vertical stripes, horizontal stripes, and squares) at three different size conditions (1, 5, and 10 mm). Each animal was tested once in a rectangular arena, which had a different pattern of the same size condition on the walls of either half of the arena. We show that zebrafish have differential preferences for patterned stimuli at each of the three size conditions. These results suggest that zebrafish have naïve preferences that should be carefully considered when testing zebrafish in paradigms using visual stimuli.

## INTRODUCTION

The zebrafish (*Danio rerio*) has emerged as a popular model organism in biomedical research and is regularly used to investigate the causes and treatments of a range of human disorders (*Kalueff, Stewart & Gerlai, 2014*). In contrast to other commonly used laboratory vertebrates, zebrafish possess many desirable qualities such as high fecundity, external fertilization, and transparent embryos (*Panula et al., 2010*). In addition to convenience, many zebrafish genes have human homologues and zebrafish brains have the same basic central nervous system regions as mammalian brains, including a significant overlap in neurotransmitter types and distribution (for review see, *Panula et al., 2010*; *Kalueff, Stewart & Gerlai, 2014*; *Stewart et al., 2015*). There are many procedures for manipulating the zebrafish genome and brain development, as well as a large variety of genetic and molecular techniques for the experimental manipulation of zebrafish (*Panula et al., 2010*; *Stewart et al., 2015*).

Reliable paradigms are necessary to detect changes in behaviour that result from genetic mutation or testing of any pharmacological or toxicological substance. There are a variety of existing behavioural tests in adult zebrafish including the T-maze, plus maze, spatial alternation task, light/dark test, novel object recognition test, episodic-like memory test,

Corresponding author
Melike P. Schalomon,
schalomonm@macewan.ca

novel tank diving test, novel approach test, and conditioned place preference tests (*Norton & Bally-Cuif, 2010*; *Kalueff, Stewart & Gerlai, 2014*; *Hamilton et al., 2016*). Some of these paradigms may require the fish to associate coloured stimuli (blue, red, green, or purple) with food rewards (*Williams, White & Messer, 2002*; *Colwill et al., 2005*). As suggested by previous research, zebrafish can have preexisting preferences for the colour of experimental stimuli, which can lead to potentially incorrect interpretation of data. For instance, one study suggested that zebrafish show an innate preference for low wavelength colours (blue and purple) compared to higher wavelength colours (red and green) (*Colwill et al., 2005*), while another study showed a preference for blue and green environments compared to yellow or red ones (*Oliveira et al., 2015*). Other researchers have found that zebrafish tend to avoid blue, prefer both red and green equally, and show intermediate preference for yellow dependent on what colour it is paired with (*Avdesh et al., 2012*). In contrast, zebrafish that were conditioned to coloured food demonstrated a strong preference for red regardless of which colour the fish had been originally conditioned to (*Spence & Smith, 2008*). Furthermore, zebrafish do not demonstrate naïve preferences for complex multicoloured stimuli (*May et al., 2016*). These studies reflect the complexity of zebrafish colour preferences and the fact that an understanding of these preferences is necessary prior to the use of behavioural paradigms involving coloured stimuli.

A simple alternative to coloured stimuli is the use of achromatic black and white patterns. Innate preferences for achromatic patterns have been tested in various model organisms (*Cunningham, Gremel & Groblewski, 2006*) but, to date, there have been no studies examining the naïve preference of adult zebrafish for basic achromatic pattern stimuli. Other organisms, such as the honeybee, have innate preferences for patterns that confer an evolutionary advantage (*Lehrer et al., 1995*). Knowledge about innate pattern preferences in zebrafish would allow the development of paradigms that can be used across different laboratories and fish populations regardless of environmental differences and is necessary to avoid bias in the testing of complex behaviours that use patterned stimuli (*Bilotta et al., 2005*; *Tierney, 2011*). In this study we tested zebrafish preference for black and white horizontal, vertical, or square patterns at three different sizes of line thickness (1 mm, 5 mm, and 10 mm).

## MATERIALS & METHODS

### Animals and housing

Adult wild-type short-fin zebrafish (*Danio rerio*) were obtained from a local supplier (Big Al's, Edmonton, Canada) ($n = 198$) and housed in 3-L or 10-L polypropylene tanks (AHAB; Aquatic Ecosystems, Inc. Apopka, FL, USA). All fish were between 1–2 years of age during the study and were housed in the habitat for at least 60 days prior to testing. The aquatic habitat was maintained at a water temperature of 26–28 °C, and a pH between $7.0 - 8.0$ as previously described (*Holcombe et al., 2013*; *May et al., 2016*). Zebrafish were fed an alternating schedule of either freeze-dried shrimp (Omega One Freeze Dried Mysis Shrimp Nutri–treat; OmegaSea Ltd., Germany) or commercial flake fish food (New Life Spectrum Optimum Fresh H$_2$O Flakes; New Life International

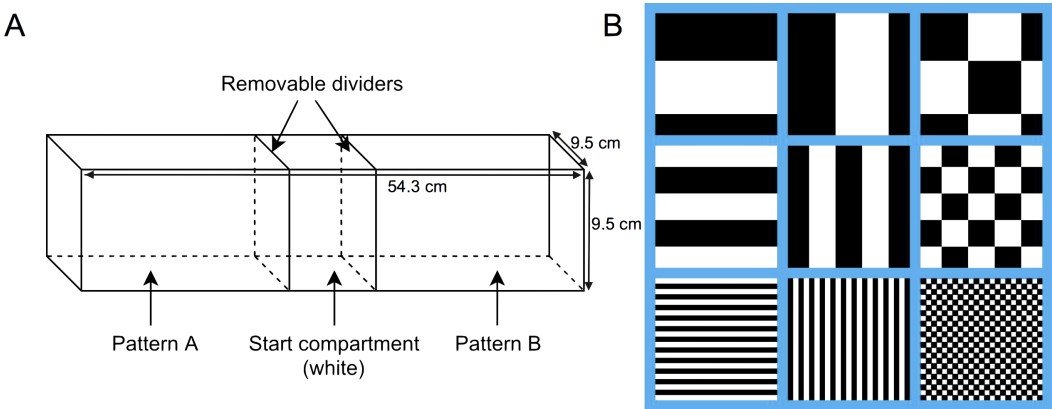

**Figure 1  Behavioural testing apparatus.** (A) Schematic of the testing tank. The whole tank (unsegregated) measured 54.3 cm long × 9.5 cm wide × 9.5 cm tall. Clear acrylic glass removable dividers were used to divide the tank into three compartments, two arms (22.5 cm long) and one center compartment (9.3 cm long), prior to the start of the trial. (B) Representative sample of each pattern (horizontal stripes, vertical stripes, and squares) for each size condition (10 mm, 5 mm, and 1 mm).

Inc. FL, USA) once per day. Lighting was provided by ceiling-mounted fluorescent light tubes on a 12-hour light/dark cycle (lights on at 8:00 am).

The experimental arena used for testing was lined with solid white, non-reflective corrugated plastic, and measured 54.3 cm long, 9.5 cm wide, and 9.5 cm tall. Clear acrylic glass removable dividers were used to divide the tank into three compartments, two arms measuring 22.5 cm long, and one center compartment measuring 9.3 cm long (Fig. 1A). The stimuli used were three different achromatic patterns in three different size conditions for a total of nine different stimuli: horizontal stripes, vertical stripes, and squares, at either 1 mm, 5 mm, or 10 mm in size (Fig. 1B). All nine stimuli were matched for luminosity, with 50% of each stimulus being black and 50% being white. Stimuli were printed on waterproof white paper with black and white patterns, and measured to fit the three walls of both arena arms. The center, or start compartment, remained lined with only the white corrugated plastic. The laminated stimulus sheets were attached to the arena arms with Velcro to permit easy switching between trials. Pattern combinations for each trial were chosen using a random number generator and the pattern locations were fully counterbalanced. The test tank was filled to a depth of 5 cm using water from the aquatic habitat. Water temperature was maintained between 26–28 °C for the duration of all trials. Fish behaviour was recorded and analyzed using Ethovision XT (v7; Noldus, Leesburg, VA, USA) motion-tracking software and a camera mounted 1m above the arena.

## Behavioural testing

Prior to each testing session, zebrafish were netted from the aquatic habitat and placed in a holding tank beside the testing apparatus until the start of the experiment $(10-60$ min$)$. Two different patterns were placed in the two arms of the tank and the dividers were put in place. At the start of each trial a zebrafish was netted and placed in the center compartment. The central acrylic glass dividers were immediately removed and the experimental trial commenced. The duration of time zebrafish spent in each of the three unsegregated
compartments for the five minute trial was recorded with Ethovision, as was swimming speed in each compartment. At the completion of each trial, the zebrafish was netted and returned to the home tank. The stimuli in each trial consisted of a pattern of a particular size condition in one arm of the arena, and a different pattern of the same size condition in the other arm. Following the completion of 198 trials, the data were exported from Ethovision XT to Microsoft Excel and then analyzed using SPSS software. There were 66 zebrafish tested in each of the 1 mm, 5 mm, and 10 mm conditions. Each fish was tested for naïve preference and therefore was only tested once, resulting in each fish being tested for preference in the presence of two stimuli (for a given size).

This research was approved by the Grant MacEwan University Animal Research Ethics Board, protocol number 05–12–13 and is in accordance with the Canadian Council for Animal Care (CCAC) guidelines.

## Statistical analyses

Data was analyzed with $F$ tests as a mixed time (or velocity) within each zone containing a patterned stimuli by block (stimuli present during the trial, i.e., horizontal stripes with vertical stripes, horizontal stripes with squares, or vertical stripes with squares) and object (pattern, i.e., horizontal stripes, vertical stripes, or squares) design. Object (pattern), block (stimuli combination), and the object by block interaction were treated as fixed effects. The object by block interaction was analyzed to determine whether there was a specific preference for a pattern (ex. vertical stripes) that was more pronounced in a given trial (ex. vertical stripes vs. squares) than the other comparison trial (ex. vertical stripes vs. horizontal stripes). Pairwise comparisons between patterns were made with a Bonferroni adjustment for multiple comparisons. Eleven trials were omitted from the statistical analysis on the basis of tracking errors by Ethovision software ($n = 6$) or because the zebrafish failed to explore both sides of the test arena ($n = 5$). Tracking errors were defined as an inability of the software system to recognize the fish relative to the background picture as a result of the fish being too small or low in contrast relative to the background. The final number of successful trials was 63 in the 1 mm condition, 64 in the 5 mm condition, and 60 in the 10 mm condition.

## RESULTS

We analyzed the data for effects of stimulus combination within each of the three size conditions by analyzing the amount of time spent in the portion of the arena lined with that particular pattern, as well as velocity (cm/s) of the fish. For the 10 mm size condition, both the main effect for object ($F (2, 114) = 9.11$, $p < 0.001$), as well as the interaction of block by object ($F (1, 114) = 9.21$, $p = 0.003$) were significant (Fig. 2A and Fig. S1A). Zebrafish spent significantly more time in the compartment with vertical stripes (164 s; SE = 9.94) than in compartments with horizontal stripes (120 s; SE = 9.69) or squares (119 s; SE = 9.82). The interaction effect reflects the increased preference for the vertical stripe pattern when it was paired with the square pattern relative to when the vertical stripe pattern was paired with the horizontal stripe pattern. There was no significant main effect for block ($F (2, 114) = 2.58$, $p = 0.08$). The velocity analysis showed no significant

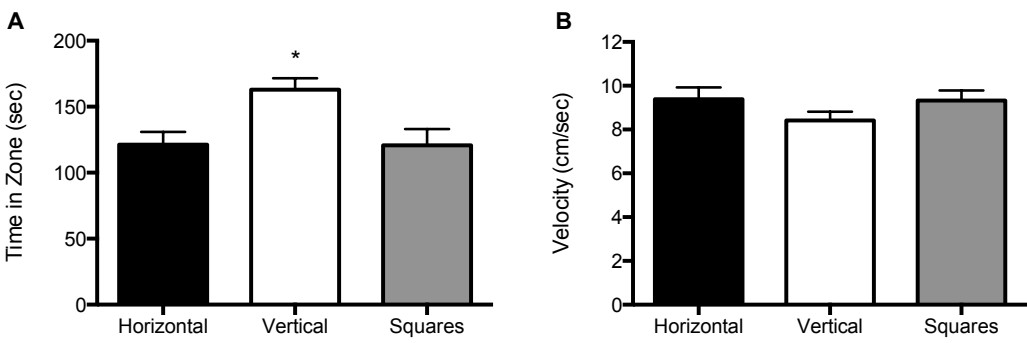

**Figure 2  Time in zone and swim speed for 10 mm condition.** (A) Time spent in the zone with the indicated pattern (mean ± SEM). Fish spent significantly more time in the zone with the vertical stripes ($p < 0.001$). (B) Swim speed as measured by velocity (mean ± SEM) of the fish within each of the pattern types. The swim speed of the fish did not differ between the three patterns.

main effects for block ($F_{(2, 114)} = 0.64$, $p = 0.53$) or object ($F_{(2, 114)} = 0.15$, $p = 0.86$; Fig. 2B, and Fig. S1B). There was also no significant block by object interaction effect ($F_{(2, 114)} = 0.047$, $p = 0.83$. The swim speed of the zebrafish was not significantly different in response to any of the three patterns at this size (horizontal stripes: 9.41 cm/s; vertical stripes: 8.70 cm/s; squares: 9.30 cm/s).

For the 5 mm size condition, both the main effect for block ($F_{(2, 122)} = 25.8$, $p < 0.001$) and the main effect for object ($F_{(2, 122)} = 95.2$, $p < 0.001$) were significant (Fig. 3A and Fig. S2A). Zebrafish spent significantly less time in the compartment with squares (69.4 s; SE = 6.24), whether it was paired with a compartment with vertical stripes (159 s; SE = 6.31) or with horizontal stripes (160 s; SE = 6.24). There was no significant effect for the interaction of block by object ($F_{(2, 122)} = 0.015$, $p = 0.90$). In the velocity analysis, there were significant main effects for both block ($F_{(2, 122)} = 3.91$, $p = 0.023$) and for object ($F_{(2, 122)} = 13.2$, $p < 0.001$; Fig. 3B and Fig. S2B). Zebrafish swam significantly faster in the compartment with squares (9.75 cm/s; SE = 0.29) whether the pattern was paired with vertical stripes (8.44 cm/s; SE = 0.29) or horizontal stripes (8.05 cm/s; SE = 0.29). There was no significant block by object interaction effect ($F_{(2, 122)} = 0.026$, $p = 0.87$).

For the 1 mm size condition, both the main effect for block ($F_{(2, 120)} = 5.39$, $p = 0.006$) and the main effect for object ($F_{(2, 120)} = 19.3$, $p < 0.001$) were significant (Fig. 4A and Fig. S3A). Zebrafish spent significantly more time in the compartment with horizontal stripes (155 s; SE = 6.60) regardless of whether it was paired with a compartment with vertical stripes (117 s; SE = 6.67) or with squares (109 s; SE = 6.51). There was no significant effect for the interaction of block by object ($F_{(2, 120)} = 2.31$, $p = 0.13$). The velocity analysis showed a significant main effect for object ($F_{(2, 120)} = 3.19$, $p = 0.045$; Fig. 4B and Fig. S3B). Overall, zebrafish swam significantly slower in the compartment with horizontal stripes (7.48 cm/s; SE = 0.32), as compared to the compartments with vertical stripes (9.07 cm/s; SE = 0.32) or with the square pattern (9.04 cm/s; SE = 0.31). There was no significant main effect for block ($F_{(2, 120)} = 2.12$, $p = 0.12$) or for the block by object interaction ($F_{(2, 120)} = 0.58$, $p = 0.45$).

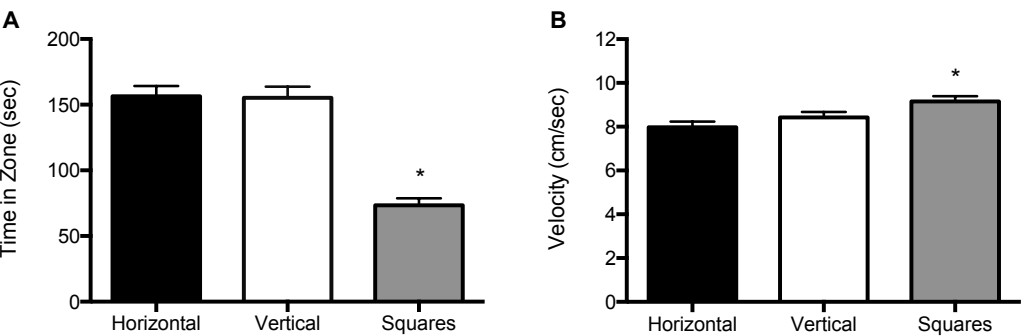

**Figure 3  Time in zone and swim speed for 5 mm condition.** (A) Time spent in the zone with the indicated pattern (mean ± SEM). Fish spent significantly less time in the zone with the square pattern ($p < 0.001$). (B) Swim speed as measured by velocity (mean ± SEM) of the fish within each of the pattern types. The fish swam significantly faster in the zone with the square pattern ($p < 0.001$).

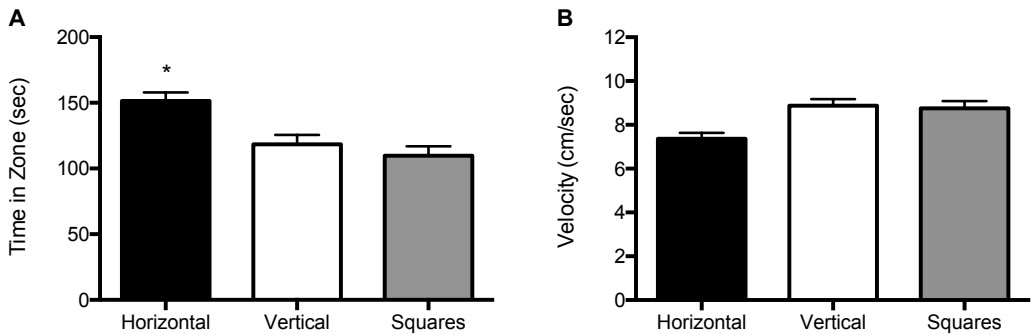

**Figure 4  Time in zone and swim speed for 1 mm condition.** (A) Time spent in the zone with the indicated pattern (mean ± SEM). Fish spent significantly more time in the zone with the horizontal stripes ($p < 0.001$). (B) Swim speed as measured by velocity (mean ± SEM) of the fish within each of the pattern types. The fish swam significantly slower in the zone with horizontal stripes ($p < 0.05$).

## DISCUSSION

Zebrafish show pattern preferences with horizontal, vertical, and square patterns at size conditions of 1 mm, 5 mm, and 10 mm, demonstrating that zebrafish have the ability to discriminate between patterns of such sizes. Vertical lines were significantly preferred at a size condition of 10 mm and horizontal lines at a size condition of 1 mm. At the intermediate size condition of 5 mm the subjects preferred neither vertical nor horizontal lines but did show a significant avoidance of the square pattern. The zebrafish also showed an increase in swim speed in response to the non-preferred 5 mm square pattern, as well as a decrease in swim speed in response to the preferred 1 mm horizontal pattern. At the 10 mm size condition there was no difference in swim speed in response to any of the three patterns.

In their natural habitat, zebrafish use vegetation as shelter from predators as well as for concealment during spawning and foraging (*Spence, Ashton & Smith, 2007*; *Spence et al., 2008*). Zebrafish have been shown to prefer environments that include vegetation in the laboratory and display less aggressive behaviours when vegetation is present (*Basquill*

& Grant, 1998; Engeszer et al., 2007b; Spence et al., 2008). Additionally, vegetation can ease the process of inflating the swim bladder in larval zebrafish, possibly conferring a survival advantage on fish spawning in vegetated areas (Spence, Ashton & Smith, 2007; Spence et al., 2008). The grasses present in a natural zebrafish environment may be visually similar to the 10 mm vertical lines preferred in this experiment.

It is plausible that the 5 mm square pattern is aversive to zebrafish and that horizontal and vertical lines of this size are both neutral stimuli. Additional evidence for the aversive nature of the 5 mm square pattern comes from the increase in swim speed, an indicator of anxiety (Kalueff et al., 2013), which occurs relative to the other patterns at this size condition. It is possible that an innate avoidance of this pattern is related to the spotted appearance of common predatory species that zebrafish encounter in their natural environment, such as the blotched snakehead (Channa maculata), northern snakehead (Channa argus), or Gangetic leaffish (Nandus nandus) (Engeszer et al., 2007b; Gerlai, 2013).

The preference for horizontal lines of 1 mm may be explained by the horizontal stripes of a similar width that characterize the appearance of conspecifics. This would be consistent with the lack of horizontal preference at thicker line widths. Zebrafish tend to form a preference for stripes that is reliant on early learning experiences during a critical period that result from an imprinting-like phenomenon (Engeszer, Ryan & Parichy, 2004; Rosenthal & Ryan, 2005; Engeszer et al., 2007a). The zebrafish used in this study were likely housed with wild-type (striped) fish from hatching and are therefore likely to express a strong preference for the 1 mm wide horizontal stripe patterns characteristic of conspecifics. Zebrafish stripes play a significant role in shoaling behaviour, as well as in the mediation of aggressive and mating behaviours between conspecifics (Rosenthal & Ryan, 2005; Engeszer et al., 2007a). Therefore, zebrafish that were raised with phenotypically similar tank mates can be expected to show a preference for a 1 mm horizontal striped pattern. In addition to a preference for the horizontal pattern, the zebrafish in this study also showed a decrease in swim speed in the compartment with the horizontal stripes at the 1 mm size but not at 5 mm or 10 mm. High swim speed is indicative of anxiety in zebrafish (Kalueff et al., 2013), thus the decrease in swim speed in response to the horizontal stripe pattern may indicate a decreased stress response. To date there have been many studies on location preference of zebrafish in the presence of various colour stimuli (Avdesh et al., 2012; Bault, Peterson & Freeman, 2015; Oliveira et al., 2015) but few have examined mobility within the stimulus compartment in a two choice test. However, when the walls of an arena are either black, white, or transparent there is a significant effect on immobility which increases with transparent walls (Blaser & Rosemberg, 2012). Indeed, the duration of time spent in a zone containing a stimulus is informative, but so is the level of movement within that compartment itself, which is why the speed (or immobility) of the fish may also be of interest. Future studies with anxiolytic or anxiogenic drugs could investigate this in more detail.

This experiment demonstrates that adult zebrafish have the ability to differentiate between achromatic patterns ranging from 1 mm to 10 mm in size. Knowledge of visual acuity in zebrafish is relevant for behavioural paradigms that rely on the perceptual abilities of zebrafish, such as ones that use patterned test tanks or object discrimination tasks. The

results in this study suggest that zebrafish have naïve preferences for some patterns in their surroundings, and they suggest that the visual acuity of zebrafish permit them to perceive patterns with a resolution of 1 mm. There were no relative preferences for the 10 mm horizontal lines or squares, the 5 mm horizontal or vertical lines, and the 1 mm squares and vertical lines, suggesting that these patterns would be suitable for use in place preference paradigms. The significance of the preferences for the pattern sizes is further substantiated by the fact that the zebrafish used in this study had diverse experimental and genetic backgrounds. However, as this is the first study to examine preferences for patterned environments in zebrafish, more testing would be beneficial to confirm the results across multiple strains. Additionally, future studies may benefit from analyzing pattern preference results by sex, as recent studies suggest zebrafish behaviour varies between males and females (*Tran & Gerlai, 2013*; *Lucon-Xiccato & Bisazza, 2017*). As zebrafish continue to gain popularity as a model organism in the study of vertebrate disease, the development of reliable behavioural paradigms will be increasingly crucial. Knowing which visual stimuli are preferred or avoided by naïve subjects is key in the development of these paradigms in order to reduce the possibility of biased or invalid data.

## ACKNOWLEDGEMENTS

We would like to thank Karen Buro (MacEwan University) for her assistance with the statistical analysis. We would also like to thank Jay Abbott (Animal Care Technician, MacEwan University) for his assistance with animal husbandry.

### Funding

This publication was supported by the Natural Sciences and Engineering Research Council (NSERC) Discovery grant to TJH (04843) and the MacEwan Research Office. The funders had no role in study design, data collection and analysis, decision to publish, or preparation of the manuscript.

### Grant Disclosures

The following grant information was disclosed by the authors:
Natural Sciences and Engineering Research Council (NSERC): 04843.
MacEwan Research Office.

### Competing Interests

The authors declare there are no competing interests.

### Author Contributions

- Lisa A. Rimstad performed the experiments, analyzed the data, wrote the paper, prepared figures and/or tables, reviewed drafts of the paper.
- Adam Holcombe performed the experiments, analyzed the data, wrote the paper, reviewed drafts of the paper.

- Alicia Pope performed the experiments, analyzed the data.
- Trevor J. Hamilton and Melike P. Schalomon conceived and designed the experiments, contributed reagents/materials/analysis tools, wrote the paper, prepared figures and/or tables, reviewed drafts of the paper.

### Animal Ethics

The following information was supplied relating to ethical approvals (i.e., approving body and any reference numbers):

This research was approved by the Grant MacEwan University Animal Research Ethics Board, protocol number 05-12-13 and is in accordance with the Canadian Council for Animal Care (CCAC) guidelines.

### Data Availability

The raw data has been uploaded as a Supplemental File.

### Supplemental Information

Supplemental information for this article can be found online at http://dx.doi.org/10.7717/peerj.3748#supplemental-information.

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
