# Peer review of "Preferences for achromatic horizontal, vertical, and square patterns in zebrafish (Danio rerio)"

_PeerJ, doi:10.7717/peerj.3748_

## Round 0.1 · original submission · Minor Revisions

Both reviewers have sensible and useful suggestions for improving the paper, and so I would encourage you to engage with all of the suggestions, even if you argue against some suggested changes (or explain why they are not possible).

Reviewer 1 ·

Basic reporting

In this manuscript the authors detail a study in which the preferences for stripe and square patterns were assessed in adult zebrafish. Overall the manuscript is well-written.

Experimental design

The experimental design is well-described and performed within ethical standards. Most details are present in the methods, except do the authors know the age of the adult zebrafish used in their experiments? This detail is important to ascertain if a younger, middle-aged, or older adult were used in this study.

Validity of the findings

The findings are valid as currently presented if ignoring the potential for sex-specific effects. Did the authors note the sex of the fish during the trials and can the analysis be conducted by sex? This is important as some researchers do report sex-specific preferences in various vertebrate models including the zebrafish. A sex-specific analysis would greatly enhance the reported findings.

Reviewer 2 ·

Basic reporting

The manuscript by Rimstad et al. describes achromatic preferences by adult zebrafish. They show that, depending on size, adult zebrafish show a preference for horizontal and/or vertical stripes, relative to a squared pattern. The study seems well designed and carried out and the findings are novel. In general, more clarity is needed regarding the statistical approach.

Experimental design

The section on the statistical analyses was very difficult to understand. Please rewrite lines 117-119, breaking the text up into multiple, simple sentences. Give examples of the comparisons made. For example, define a block using the terminology shown in the figures (combination of data derived from horizontal, vertical and square pattern conditions). I am still not exactly sure what an object by block interaction is. The statistical strategy is key to understanding this data set. The authors should endeavor to make the strategy clear and easy to interpret to non-statisticians.

Related to the previous point, please include a single table with all of the statistical results in the revised manuscript.

Please detail what “tracking errors” means (line 121). In general, the tracking software should be pre-calibrated such that all animals are reproducibly tracked.

Validity of the findings

No comment.

Additional comments

In reference to figure 2A (10 mm pattern), line 131 says that “the interaction effect reflects the increased preference for the vertical stripe pattern when it was paired with the square pattern”. If you look at figure 2A, the data shown there suggests that there is a preference for vertical stripes relative to both horizontal and square patterns. Is there a better way to visually represent the data that would more accurately reflect pairwise comparisons?

Related to the previous point, the authors should consider showing the data as box and whisker plots with overlaid scatter data showing individual data points. The study was well designed and powered. Readers would benefit from seeing the data in a way that more shows mean, error, and individual data points.

In Figure 4, it intuitively makes sense that the animals that spend more time in the horizontal patterned side swim at a slower velocity. The discussion section should give some interpretation of when this is not the case (as shown in figures 2-3). Are there examples in the chromatic literature where time spent on one side increases with a concomitant decrease in swimming speed and other examples where that is not the case?

Remove line 170-172, regarding a non-significant trend.

Minor comments
1. Please cite primary literature in the introduction, rather than review articles (e.g. line 35).
2. Include the pattern size (10 mm, 5 mm, 1 mm) at the top of each graph for easy reference.
3. Include dimensions in figure 1A.

---

## Round 0.2 · accepted · Accept

A good, clear job has been done dealing with the suggestions, and so I am happy to accept the revised version of the MS.